# A Tale of Four Danish Cities: *Legionella pneumophila* Diversity in Domestic Hot Water and Spatial Variations in Disease Incidence

**DOI:** 10.3390/ijerph19052530

**Published:** 2022-02-22

**Authors:** Søren A. Uldum, Lars G. Schjoldager, Sharmin Baig, Kelsie Cassell

**Affiliations:** 1Department of Bacteria, Parasites and Fungi, Statens Serum Institut, 2300 Copenhagen S, Denmark; basj@ssi.dk; 2CheckPoint World, 8800 Viborg, Denmark; checkpointworld@gmail.com; 3Epidemiology of Microbial Diseases, Yale University School of Public Health, New Haven, CT 06510, USA; kelsie.cassell@yale.edu

**Keywords:** domestic hot water, *Legionella pneumophila*, *Legionella* colonization, Legionnaires’ disease, serogroup, sequence-type, whole genome sequencing, phylogenetic relatedness

## Abstract

Denmark has one of the highest Legionnaires’ disease notification rates within Europe, averaging 4.7 cases per 100,000 population annually (2017 to 2020). The relatively high incidence of disease is not uniform across the country, and approximately 70% of all domestically acquired cases in Denmark are caused by *Legionella pneumophila* (LP) strains that are considered less virulent. The aim of this study was to investigate if colonization rates, levels of colonization, and/or types of LP present in hot water systems were associated with geographic differences in Legionnaires’ disease incidence. Domestic water systems from four cities in Denmark were analyzed via culture and qPCR. Serogrouping and sequence typing was performed on randomly selected isolates. Single nucleotide polymorphism was used to identify clonal relationship among isolates from the four cities. The results revealed a high LP colonization rate from 68% to 87.5% among systems, composed primarily of non-serogroup 1. LP serogroup 1 reacting with the monoclonal antibody (MAb) 3/1 was not identified in any of the systems tested, while MAb 3/1 negative serogroup 1 strains were isolated from 10 systems (9.6%). We hypothesize that a combination of factors influences the incidence rate of LD in each city, including sequence type and serogroup distribution, colonization rate, concentration of *Legionella* in Pre-flush and Flush samples, and potentially building characteristics such as water temperature measured at the point of use.

## 1. Introduction

*Legionella* spp. are environmental bacteria often found in the humid settings of both natural and man-made water systems and are aerosolized through sources such as showers, fountains, mist machines, spa pools, and cooling towers [1]. Once aerosolized and inhaled, the bacteria can cause a severe pneumonia called Legionnaires´ disease (LD). The notification rate of LD in Denmark (DK) has increased from approximately two cases per 100,000 inhabitants per year between 2006 and 2013 to >4.5 per 100,000 between 2017 and 2020. Therefore, DK has a high incidence of LD when compared to other countries reporting to the European Centre for Disease Prevention and Control (ECDC) [2,3,4,5]. This upward trend in incidence over time is present in several other European countries (e.g., Italy, Slovenia, Czech Republic, and Belgium) where the average annual notification has approximately doubled between 2011 and 2018 [2,4]. The incidence of domestically acquired LD varies greatly from province to province in DK, ranging from 1.6 to 9.6 cases per 100,000 inhabitants in 2019, and the reason for geographic variation is unknown [3,6,7].

The majority of cases (>95%) reported to the ECDC are caused by the species *Legionella pneumophila* (LP) which can be divided into 16 serogroups, of which serogroup 1 is reported as the most prevalent (>85%) among clinical cases [2,4]. Serogroup 1 isolates can further be divided into subgroups, and all LP isolates can be categorized into Sequence Types (STs), with some serogroups and STs more commonly found in the environment and others more commonly attributed to clinical disease. Denmark has a relatively high frequency of clinical cases caused by LP strains that, in other countries, are more commonly associated with the environment than with cases. These strains of unusual occurrence among Danish cases belong to LP non-serogroup 1 (non-SG1) and serogroup 1 (SG1) strains that are negative for the virulence marker detected by the monoclonal antibody (MAb) 3/1 (e.g., SG1 subgroup OLDA/Oxford) [8]. The MAb 3/1 recognize an LPS epitope encoded by a functional *lag-1* gene (an O-acetyltransferase) [9]. In 2015, 80% of SG1 cases in Europe were of the virulent MAb 3/1 positive (MAb 3/1-P) group, compared to only 50% of domestic SG1 cases in DK [3,4,7]. Among all Danish cases, including travel-associated cases, around 60% are due to SG1 (30% MAb 3/1-P and 30% MAb 3/1 negative (MAb 3/1-N)), while LP serogroup 3 accounts for 20–25% of all cases, and other non-SG1 for 15–20% of the remaining cases [3]. The less-virulent strains are especially prevalent among nosocomial and healthcare-associated cases, consistent with reports from elsewhere in the E.U., but this group accounts for less than 10% of domestic cases in DK [8,10,11,12,13,14].

A high proportion (>95%) of Danish domestic cases are sporadic, without an identifiable common source. It is assumed that most cases are infected via domestic hot water systems (DHW), although this can only be documented for approximately 50% of culture-confirmed cases when domestic water samples have been collected [3,7]. The subgroup OLDA/Oxford accounts for 50% of all environmental SG1 isolates in Denmark, and the vast majority belongs to ST1. ST1 is also the most prevalent ST isolated from environmental samples in many other parts of the world [13,15,16]. Serogroup, subgroup, and sequence type classification of *Legionella* isolates are important to understanding the risk of human infection and potential sources of environmental exposure [17]. The environmental distribution of *Legionella*, species, serogroups, and sequence types does not reflect the distribution among clinical cases (this may, in part, be due to the use of *L. pneumophila* urinary antigen tests (UAT) which only detect LP SG1 (both MAb 3/1-P and MAb 3/1-N) [13,18,19]. In Denmark, polymerase chain reaction (PCR) is the primarily diagnostic tool (80–90% of cases), followed by culture of lower respiratory tract samples (~39% cases each year are culture confirmed), which provides robust ST and SG data for epidemiological studies and investigation [2,4,20].

To assess whether the regional differences in LD incidence and unusual patterns in ST and sero/subgroup distribution identified among Danish LD cases correlated with the presence and ST/SG detected in domestic hot water systems, we collected DHW samples from multiple residential buildings from four major Danish cities, two with high incidence ant two with low incidence of LD (approximately 10 and 2 cases per 100,000 per year, respectively). We describe the distribution of *Legionella* SG and ST, as well as characterize the association between *Legionella* presence/concentration and building level characteristics, irrespective of plumbing structure and drinking water quality markers that are known to influence *Legionella* growth.

## 2. Material and Methods

### 2.1. Culture-Confirmed LD Case Assessment in Four Danish Cities

A retrospective study of LD cases from four selected cities in DK between 2010 and 2020 was included for case assessment. Each case of LD in Denmark is registered in The Danish Legionnaires’ Disease Surveillance Register by place of residence (ZIP code) and categorized according to most probable source of infection (community-acquired, travel-associated, or healthcare-associated). Case laboratory results including sero- and DNA-typing results for each isolate are provided. All cases diagnosed with LD from the ZIP codes of interest (corresponding to the four cities selected) were extracted from the surveillance register. Travel-associated cases (both domestic and international) were excluded. For the cities (“A” and “B”) with high incidences, cases from 2014 to 2020 were extracted, and from the cities with low incidence (“C” and “D”) cases from 2010 to 2020. Eleven years of case data was selected from the two low-incidence cities in order to increase the sample size (Table 1).

### 2.2. Samples from Hot Water Systems

Two cities (A and B) with high incidences of domestic LD cases (10 cases per 100,000 per year) and two (C and D) with low incidences (<2.5 cases per 100,000 per year) were selected. Between 24 and 30 residential buildings were selected for Domestic Hot Water (DHW) sampling in each of the four cities of interest (104 buildings in total). Buildings were selected to represent a wide range of sizes (minimum seven flats per building), age, and geographic distribution across the cities. None of the residential buildings sampled were associated with known cases of LD in the past. Samples were collected during the period from 29 June 2020 to 24 September 2020. Two samples of one liter each were collected from each residence via the hot water tap (bathroom or kitchen). The first liter collected was a “Pre-flush sample” and the second liter was collected after flushing for one minute. Water temperature was measured after one minute during the collection of the “Flush sample”. The samples were sent to Statens Serum Institut (SSI) at ambient temperature for culture and qPCR analysis; all were received and processed within 24 h.

### 2.3. Characterization of Residential Buildings and Hot Water Systems

The buildings were characterized according to size (total number of flats), year of construction, and whether the system was operated with or without a hot water tank (boiler). The hot water systems were further characterized by the measured hot water tap temperatures after one minute of flush. All buildings were heated by district heating, which is common among Danish cities, given 63% of all Danish households are supplied with district heating [21]. The Danish cities included in this study are supplied by untreated groundwater (i.e., no chlorination), and none of the cities shared the same groundwater source. Additionally, none of the buildings included in this study treated the water within their premises. Data on plumbing material and water quality indicators (e.g., organic matter, pH, heterotrophic plate count, protozoans, iron, water hardness, etc.) that could influence the growth of *Legionella* were not collected as a part of this study.

### 2.4. Culture of Legionella from Water Samples and Sero-/Subgroup Determination

All samples collected were analyzed by culture, and samples from alternating residences (e.g., first, third, fifth, etc.) enrolled in the study were also analyzed via qPCR (after dividing the liter collected into two). Culture of *Legionella* was in accordance with ISO 11731:2017. In short, 2 × 500 µL were plated directly on two GVPC (Oxoid, Thermo Fisher Diagnostics) agar plates, the rest were filtered through a 0.22 µm polyethersulfone membrane filter (MicroFunnel Plus, Pall Life Science), and each filter was vortexed with glass-beads for four min. with 10 mL of the sample. From each filtrate, 2 × 100 and 2 × 500 µL were seeded to GVPC agar plates. The plates were incubated at 36 °C in plastic bags for seven days. The plates were inspected after two days, and if growth of interfering bacteria were observed (less than 5% of samples), aliquots of the filtrate (kept at 4–10 °C) were heat- and acid-treated and 2 × 100 µL of each were plated on GVPC agar plates and incubated for seven days. The plates were inspected and three *Legionella* colonies (if present) from each sample were analyzed with the Oxoid Latex Agglutination Test (Oxoid, Thermo Fisher Diagnostics). This test provides separate identification of LP SG1, SG 2-14 (non-SG1), and detection of other *Legionella* species. In the case of doubt, colonies were analyzed by MALDI-TOF [22]. Subgroup determination was made for all colonies identified as SG1 by the Oxoid Test. At least one non-SG1 LP isolate (if present) was randomly selected (with five exceptions) from each system for serogroup determination by ELISA with the Dresden Panel of monoclonal antibodies [8]. In total, 103 isolates (28 from City A, 29 from City B, 26 from City C, and 20 from City D) were sub/serogrouped. All detected *Legionella* non-*pneumophila* species (detected in nine systems) were identified to the species level by MALDI-TOF. *Legionella* colonies were counted for each plate, and the highest colony count among the three steps was reported as the result and expressed as a concentration in colony-forming units per liter (CFU/L). Direct plating was used for enumeration only when ≥ 5 colonies were identified in total on the two plates. The limit of detection by the culture method is 100 CFU/L.

### 2.5. Whole-Genome Sequencing, Sequence Type Determination, and SNP Analysis

Whole-genome sequencing (WGS) was performed on extracted genomic DNA. Libraries were generated using Nextera XT Kit (Illumina) and 2 × 150 bp kit on NextSeq 550 (Illumina) was used for sequencing. The sequence types (STs) were determined according to ESCMID *Legionella* Study Group (ESGLI) Sequence-Based Typing (SBT) scheme [23] and were extracted from the whole-genome reads (WGS/SBT). In cases of non-identical copies of the *mompS* gene, the *mompS* tool described by Gordon et al. in 2017 was applied [24]. Altogether, 73 isolates, 15 to 21 from each city (19 from City A, 18 from City B, 21 from City C and 15 from City D), were randomly selected for WGS/SBT among isolates with known sero-/subgroup. Clinical isolates obtained before 2017 (Table 2) were analyzed by traditional SBT [23].

Phylogenetic relatedness among the STs detected in DHW systems of the four cities was analyzed by using NASP v. 1.2.0 [25] with BWA-NEM, and subsequent single nucleotide polymorphisms (SNPs) were called using GATK. The Illumina reads were aligned against the reference *L. pneumophila* strain Paris (ST1) (GenBank accession no. NC 006369.1). Recombinant regions were removed using Gubbins v. 2.3.4. The relatedness of the isolates was inferred using IQ-TREE v. 2.0.3 [26] and visualized in iTOL.

### 2.6. qPCR for Legionella Spp., L. pneumophila, and L. pneumophila Serogroup 1

Quantitative polymerase chain reaction (qPCR) for *Legionella* spp., LP, and LP SG1 were performed on samples (1/2 L) from every other building (53 systems, 106 samples). The method as well as a summary of the results and a short discussion are presented in the Appendix A.

### 2.7. Assessing the Association with Building Characteristics

Differences in the concentration of *Legionella* (log-transformed CFU/L) in Pre-flush and Flush samples between cities were assessed by an ANOVA. To identify predictors of *Legionella* concentration (log-transformed CFU/L) in Pre-flush and Flush samples, simple linear regressions were fitted (Appendix A). The outcome of the regression analysis was the log of *Legionella* (CFU/L), and predictors were building characteristics of interest: temperature (°C), building age (years), building size (No. of flats), hot water tank presence (yes/no), and city (reference value: City A). Buildings were categorized as having <45 flats or ≥45 flats, as 45 was the average number of flats among buildings sampled in this study. Separate analyses were conducted for Pre-flush and Flush samples. Similar models were fitted to the data collected from each city individually in order to investigate city-specific associations.

### 2.8. Assessing the Association between Age of Culture-Positive Cases and Infection with MAb 3/1 Positive Strains vs. MAb 3/1 Negative and Non-SG1 Strains

The only risk factor known for the cases from the four cities is age. The age distribution for cases with a culture of LP MAb 3/1-P strains (*n* = 29) was compared with the age distribution of cases with a culture of MAb 3/1-N or non-SG1 strains (*n* = 63) by an ANOVA.

## 3. Results

### 3.1. Incidences of LD Cases and Culture-Confirmed LD Cases in the Four Cities and Types Detected in Clinical Samples

In total, 191 LD cases (92 culture-confirmed) were identified in the four cities of interest between 2010 and 2020 (Table 1 and Table 2). Cases occurring in cities B and D were of older ages (74 and 73, respectively) than the median of all cases in Denmark in 2018 and 2019 (median age: 69) (Table 1) [3]. City A had an uncommon and unexplained low male to female ratio approximating 1:1 for both total number of cases and for culture-confirmed cases. The LP ST distribution among clinical samples ranged from 23 different STs isolated from City B cases to only five different STs among City D cases (Table 2). SG1 MAb 3/1-P subgroup comprised 31% of the culture-confirmed LD cases in these four cities, followed by SG3 (26%-of cases) and SG1 MAb 3/1-N (20% of cases). The overall sero/subgroup distribution for clinical isolates from the four cities is shown in Figure 1.

The mean age for MAb 3/1-P cases was 63 years, and the mean age for MAb 3/1-N and non-SG1 cases was 72 years. The difference in age distribution was highly significant by ANOVA (*p* < 0.001). The male-to-female ratios were 1:1.71 and 1:1.25 for the MAb 3/1-P group and the MAb 3/1-N and no-SG1 group, respectively.

### 3.2. Culture of Water Samples from the Four Cities—Colonization Rate and Concentrations of L. pneumophila

The colonization rates of LP (systems culture-positive of systems investigated) and the concentrations of cultivable *Legionella* for Pre-flush and Flush samples are presented as the median, mean, and standard deviation (SD) values in Table 3. Only systems with growth of LP are included, as no cases of *Legionella* non-LP were detected in the cities (Table 2). A greater concentration of *Legionella* (CFU/L) was present in Pre-flush samples compared to Flush samples among all cities assessed. Cities A and B had positivity LP rates above 80% (87.5% and 83.3% respectively), whereas Cities C and D had positive rates of 72% and 68%, respectively; however, the differences between these high and low incidence cities were not statistically different (Chi-square *p*-value: 0.20). In Table 4, the levels for Pre-flush and Flush samples are indicated as proportions of samples with *Legionella* (CFU/L) within the indicated ranges.

### 3.3. Legionella Species and L. pneumophila Sero-/Subgroup/Sequence Type Distribution among DHW in the Four Cities

Altogether, 11 different LP sero/subgroups were identified among 103 isolates from the DHW samples. In Figure 2, the relative proportions of *Legionella* species and LP sero/subgroups detected (and systems with no growth) are presented. In general, one to two colonies from each culture positive DHW system were identified down to species and sero/subgroup (Figure 2). LP SG1 of the MAb 3/1-P group was not identified in any of the systems (Figure 2). SG1 MAb 3/1-N was only isolated from 10 systems (9.6%) among cities A, B, and D (0% in City C, 4% in City D, 12.5% in City A, and 20% in City B). The LP sero/subgroup distribution varied greatly, with samples from each city revealing a unique composition (Figure 2).

Altogether, 26 different STs were identified among the 73 investigated LP isolates from the DHW systems and 80.8% of the STs were detected in one City only (21/26). The diversity of STs (and associated sero/subgroups) per system for each city is shown in Figure 3 (70 systems in total). ST87 was the only ST detected in all four cities and was most prevalent in City A (53% of isolates with known ST) (Figure 3). SG1 ST1 was only isolated in City B. ST80 (SG5 Cambridge) and ST1333 (SG4/SG10) were the second most commonly isolated ST, each being isolated from six samples.

### 3.4. Single Nucleotide Polymorphism (SNP) Analysis

Phylogenetic trees of isolates (*n* = 77; 73 plus four replicates) from the DHW systems were constructed based on SNP analysis, and the results are presented in Appendix A. Appendix A shows a phylogenetic tree with recombinant regions (unpurged) which is based on 143,204 SNPs. Appendix A, with the removal of recombinant regions (purged), is based on 97,458 SNPs. The isolates cluster in three major clades (A, B, and C). Most isolates (*n* = 46) are in Clade A, where ST87 (SG3 for 17 of 20 isolates) is the main ST (SBT profile 2,10,3,28,9,4,13). Several STs are in this cluster, including ST371 and ST338 (both SG3), which are single locus variants (SLVs) of ST87, and are within the ST87 lineage. SG3 was mainly associated with the ST87 lineage, with one exception being ST499. Other isolates (STs) within this clade (with two to five loci differences relative to ST87) constitute small sub-clusters (Appendix A). Strains belonging to Clade A were detected in all four cities. Clade C is phylogenetically remotely related to Clade A (Appendix A) with a minimum of ~50,000 and ~7000 SNPs, respectively, for unpurged and purged results. In comparison, the maximum number of SNPs within Clade A is ~12,000 and ~1200 for unpurged and purged, respectively. Isolates in Clade C were only detected in City A, B, and C, and isolates in this clade are primarily SG3 (8 of 13 isolates). A sub-cluster in this clade was only found in City B and constitutes of ST337 (SBT profile 10,22,7,28,16,18,6) and two SLVs ST993 and ST2187 and a two loci variant ST2589. Cluster B is heterogeneous with three main sub-clusters (Appendix A); all SG1 isolates are within this cluster, including the SG1 ST1 sub-cluster; other serogroups (SG5, 6, and 10) are also found in this clade, but not SG3 (Appendix A).

### 3.5. Association between Presence and Concentration of Legionella and Building Characteristics

The mean concentration of *Legionella* in the Pre-flush samples was approximately two to nine times greater than the flush samples from the study cities. City C had the highest mean concentration of *Legionella* in the Pre-flush samples (56,144 SD: 198,643), which decreased to a mean of 6860 in the Flush samples; however, due to the large variance in sample concentrations, the greatest median concentration of *Legionella* among Pre-flush samples belonged to City B (Table 3 and Figure 4). 53% of Pre-flush samples in City B measured greater than 10,000 CFU/L of *Legionella* (Table 4). City B also measured the highest median concentration for Flush samples of 2150 CFU/L, with 33% of Flush samples having a concentration of *Legionella* > 10,000 CFU/L (Table 3 and Table 4). The mean of Pre-flush and Flush samples (log of CFU/L) did not differ significantly by city (ANOVA *p* > 0.05) (Figure 4).

The mean temperature of the DHW after one-minute flush (Flush sample) among the buildings sampled was 48 °C, and ranged from 35.4 to 60.1 °C. At temperatures above 55 °C, Flush samples had no detectable or low concentrations (≤1000 CFU/L) of cultivable *Legionella*, but only a few systems had temperatures in this range (*n* = 5). Univariable models revealed that increasing temperature was associated with decreasing *Legionella* concentration (log of CFU/L) for both Pre-flush (*p* < 0.05) and Flush (*p* < 0.05) among samples detecting *Legionella* (*n* = 78) (Figure 5). This trend remained when Pre-flush and Flush samples negative for *Legionella* presence were included (Appendix A). This association was primarily driven by samples from City B and City D.

Building age was not consistently associated with increasing *Legionella* concentration (Figure 5). When including samples with 0 CFU/L *Legionella* detected, City A was the only city that exhibited an association between increasing building age and an increase in *Legionella* concentration for both Pre-flush (*p* < 0.001) and Flush samples (*p* = 0.001) (Appendix A). Neither Pre-flush nor Flush samples (CFU/L) from buildings with ≤45 flats (*n* = 72) differed significantly in mean *Legionella* concentration from buildings with >45 flats (*n* = 32). In City B, 90% (27/30) of the systems sampled had hot water tanks (boilers), in some cases in series, whereas the majority of systems in the other cities had heat exchangers without hot water tanks (City A 100%; City C and D 88%) (Table 3).

## 4. Discussion

### 4.1. Discussion of Legionella Concentrations/Colonization Rates

High LP colonization rates were observed in all four cities, with viable LP present in 68 to 87.5% of DHW systems investigated. Between 44 (City D) and 63% (City B) of the systems from each city had levels of *Legionella* ≥ 1000 CFU/L among one-minute Flush samples. The only city to have a majority of systems (56%) with concentrations ≤1000 CFU/L of LP in Flush samples was City D. A limit of 1000 CFU/L is generally recommended as the maximum acceptable level of *Legionella* in potable water supplies, and is on the high end of what has been estimated from QMRA models assessing clinical severity infection (CSI) from common home exposure [27,28,29]. Between 12 and 33% of the systems tested had levels of >10,000 CFU/L in one-minute Flush samples, which could pose a risk for the users of the water system. At this level, most guidelines advise remedial actions including system disinfection [28,29]. Compared to reports from other countries, this colonization rate and concentration of *Legionella* appears relatively high. In general, *Legionella* can be cultured from only 20 to 30% of sampled residential hot water systems in Europe and USA [19,30,31,32,33,34,35,36]. Higher colonization rates are, however, reported for hospitals and hotels [37,38,39,40].

Pre-flush samples from City B had a median *Legionella* concentration (14,500 CFU/L) that was 2.6 to 9.0 times greater than that of the other cities; however, the mean *Legionella* concentration was similar between all cities. This underscores the limitations of cross-sectional sampling of *Legionella*, as the range in concentrations can diverge immensely. The mean temperature of the hot water systems in City B did not differ from the other cities’ in this study, and we suspect that other factors, such as the unusually high proportion of hot water systems with hot water tanks (boilers) could instead contribute to the high LP colonization rate of 87.5% and concentration of LP in this city. The greater prevalence of hot water tanks could be a risk factor for growth of *Legionella*, and is especially of concern if the hot water tanks are over-dimensioned relative to the water consumption [29,30,41]. It is, however, unknown how this could contribute to the generally high concentrations measured in the Pre-flush samples in this city. The majority of samples in this study noted a reduction in *Legionella* concentration in the flushed samples compared to the Pre-flush samples, which is consistent with other similar studies [42]. The largest reduction was noted for City B, with a 6.7 times difference between the median *Legionella* concentration for Pre-flush and Flush samples (Table 3), which could indicate a substantial growth of *Legionella* in the distal parts in many of the DHW systems in this city.

### 4.2. Discussion of SG1 MAb-3/1 Positive Samples

The overall sero/subgroup distribution among clinical isolates from the four cities shown in Figure 1 can be considered as representative for the whole country, as it approximates the country´s overall sero/subgroup distribution. SG1 MAb 3/1-P was not detected in any of the DHW systems, indicating a low prevalence of these subgroups in domestic water systems in DK, which is in accordance with the relatively low prevalence of MAb 3/1-P strains of 30% among clinical isolates. Interestingly, 10 of the 22 culture-confirmed cases detected in City A were caused by LP SG1 subgroups Philadelphia ST1 and Benidorm ST42 (MAb 3/1-P group), yet these STs were not detected in any of the 24 domestic hot water systems investigated. SG1 (MAb 3/1-P) might be a rare occurrence in City A that causes a disproportionate number of LD cases (2.1 culture-confirmed cases per 100,000 inhabitants per year). This finding may be supported by several studies demonstrating that the MAb 3/1-P strains have a higher virulence than other LP serogroups and subgroups [8,13,43,44,45]. During the last four years, the only MAb 3/1-P strains that have been detected in environmental samples (primarily DHW samples) in DK are Philadelphia ST1 and Benidorm ST42, despite 63% of domestic SG1 MAb 3/1-P cases (*n* = 94) being caused by other Benidorm, Allentown/France, Philadelphia, and Knoxville associated STs. The ecological niches for these strains could be different from strains often found in DHW installations [19].

### 4.3. Discussion of SG1 MAb 3/1 Negative and Non-SG1

Although SG 1 OLDA/Oxford ST1 (MAb 3/1-N) is the most prevalent ST among Danish LD patients, it was only detected in City B, indicating that this ST is not distributed evenly across the country, but if present, probably poses an increased risk relative to other frequently detected environmental types, and could potentially drive regional differences in LD incidence. SG1 is reported to be present in 50% to 83% of culture-positive environmental samples in some countries [13,19,32,33,46,47]. In our survey, only 9.1 (culture) to 13.6% (qPCR) of systems were colonized with SG1 (three systems, one in City D and two in City B, were probably colonized by LP SG1 only) compared to a LP non-SG1 average colonization rate of 75%. One study from Italy identified non-SG1 in 62% of samples, but this was mainly in samples from hospitals, care homes, and hotels [40]. Non-SG1 colonized 77% of the DHW systems in City B but “only” caused 46% of the culture-confirmed LD cases occurring in this city. The yearly incidence of culture-confirmed non-SG1 infections among inhabitants of City B (of 2.4 cases per 100,000) was more than three times that experienced in Cities C and D (Table 2). It is possible that the high incidence of non-SG1 infections in City B may be related to the higher colonization rate and concentrations of LP detected in this city compared to Cities C and D.

The colonization rate of the OLDA/Oxford SG1 subgroup (MAb 3/1-N) in City B was 20%, and caused a proportional number of cases (29% (Table 2)); this could indicate a higher virulence for this SG1 subtype than for non-SG1 LP in general.

SG3 was the most prevalent SG detected in the DHW systems (33/104; 31.7%), and was detected in all four cities; however, with varying prevalence (Figure 2). SG3 caused a proportional number of cases in the four cities (26%) (Figure 1). SG3 is reported as a rare cause of LD, accounting for 3% (*n* = 35) of all culture-confirmed cases in the EU/EEA in 2019 [2]; however, 25 of these reports were reported from DK. It is not known why SG3 has a high prevalence in DK, and reports from other countries rarely indicate the rate of specific SGs in environmental samples to allow for a proper comparison. Studies from Greece [48] and Israel [49] report a relatively high proportion of SG3 in culture-positive water samples, identifying SG3 in 32.5% of isolates from samples culture-positive for non-SG1 in Greece and in 60.44% of LP culture-positive samples in Israel. We cannot identify common factors between Denmark and Greece/Israel that could explain the similarities, as the climatic conditions are quite different and the public water supplies both in Greece and Israel are treated with chlorine, whereas the public water in Denmark is left untreated.

Of the 24 SG3 cases from the four cities (Table 2), it is known that the corresponding serogroup and ST were detected in DHW samples for seven of the cases (ST87 in five and ST93 in two cases). SG3 ST93 is the third most common ST (after ST1 and ST87) detected in clinical samples in Denmark and accounts for approximately 5% of cases (the same proportion as found in the four cities (Table 2)). This ST was not detected in water samples from this survey, but the ST (SBT profile, 3,10,1,28,14,9,13) is relatively closely related to ST87, sharing four common ST loci.

### 4.4. Discussion of ST

ST1 (Philadelphia and OLDA/Oxford) and ST87 account for approximately 35% of all domestic cases in Denmark each year, but with local differences in incidence that may be related to differences in DHW presence. ST87 (predominantly associated with SG3) is the second most prevalent ST among patients in DK (approximately 10%), and was found in all four cities, although not with the same high prevalence as seen in City A, where 10 of 16 LP-positive systems with known ST results were colonized with ST87 (62.5%). ST87 also had a high prevalence among clinical LD cases from this city (5/23; 22%). City C also had a relatively high prevalence of ST87 among DHW systems (5/18; 28% of systems with known ST) and was the most prevalent ST among clinical isolates from this city (3/11; 27%). It is possible that ST87 has a higher virulence than most other non-SG1 strains, but the high prevalence of this ST among clinical isolates might also be attributed to a widespread presence in the environment, although apparently not distributed evenly across the country. Six of the mentioned 10 Mab 3/1-P cases reported in City A (Table 2) had a corresponding sequence type (ST1 or ST42) detected during the case investigation (data not included in this study) in the sampled residential hot water systems, indicating that the clinical STs are also present in the DHW systems of this city, but likely sporadically. Strains such as ST23, ST37, ST47, and ST62 that typically are associated with most clinical cases in the rest of Europe and outbreaks [18,50,51,52] are rarely seen in clinical cases in DK, and have never been registered as isolated from any environmental sample in DK. We did not investigate potential external sources such as cooling towers in this study; cooling towers are, however, considered as very rare sources of LD in Denmark. Not all LD cases in DK are due to contaminated DHW systems, but clusters/outbreaks caused by external sources are rarely identified. The two cases caused by Allentown/France ST82 in City C (Table 2) could be part of a cluster caused by an external source, as the two cases resided close to each other and had disease onset one month apart. LP serogroup 1 Knoxville ST9 was the second most prevalent type detected in City B among patients (Table 2), but this is probably due to an outbreak (source unidentified) in late 2015 and at the beginning of 2016, comprising seven of the eight ST9 cases. The outbreak was most likely due to a common external source (e.g., cooling tower), as the ST was not was identified in any of the investigated households.

### 4.5. Discussion of SNP Analysis

The restricted SNP analysis only included environmental isolates; STs from the clinical isolates could be included for a more comprehensive analysis. The analysis shows, however, a widespread prevalence of strains closely related to one of the major disease-causing STs in Denmark (ST87) (Clade A), and is particularly associated with SG3 (Appendix A). ST338, ST371, ST728 (SVLs of ST87), and ST1333 (a two loci variant of ST87) have all caused human infection in the four cities (Table 2), whereas none of the other STs in the clade with ≥2 loci difference to ST87 (ST292, ST421, ST499, ST1916, ST2005, and ST2923) were identified among clinical isolates from the four cities. Among ST87 isolates from different cities, the maximum number of SNPs were 17 (purged), and the minimal distance was one SNP, indicating that the same ST87 clone has dispersed to all four cities. Clade C is a rather homogenous group when recombination events are removed (maximum 45 SNP distances) (Appendix A); SG3 is the main serogroup in this cluster (8 of 11 isolates) but phylogenetically distant from SG3 in Clade A. All SG1 isolates cluster in Clade B, including ST1 (OLDA/Oxford; SBT profile 1,4,3,1,1,1,1) and ST59 (Bellingham, SBT profile 7,6,17,3,13,11,11), which are well-known pathogens, and were responsible for LD in the respective cities where they were found (Table 2). Although in the same clade, they are not closely related, with a SNP distance of ~16,150 SNPs (purged). SG1 Bellingham ST59 is more closely related to another disease-associated clone SG5 subgroup Cambridge ST80 (SBT profile 7,6,3,8,13,11,3), sharing four loci, and only ~15 SNPs (purged) separate the two STs. ST80 was especially prevalent in City C but caused more cases in City B (*n* = 6) (Table 2). SG3 was not detected in Cluster B.

Removing the recombinant regions removes much of the heterogeneity within the clades and sub-clusters (Appendix A), supporting that recombination is a main driver for the evolution and genomic diversity of LP, as shown by David et al. in 2017 [53]. The results also show that there is no tight association between STs and serogroups. SG3 was associated with strains that were relatively distant phylogenetically, as seen for Clade A and C. On the other hand, some STs could also be associated with multiple SGs, as for ST87, ST337, ST421 and others (Figure 3, and Appendix A). Other STs are more constantly associated with an SG; ST1 is, for example, always associated with SG1. Serogroup-associated genes are both clonally distributed and horizontally acquired, as shown for the *lag*-*1* gene (MAb 3/1-P) that has been distributed horizontally across all major phylogenetic clades of *L. pneumophila* [45]. The two SNP trees (purged and unpurged) show the same overall phylogeny, and the STs (number of loci similarities and differences) are good indicators of phylogenetic relatedness.

### 4.6. Building Characteristics

This study found relatively high concentrations of LP in the hot water systems, regardless of sero/subgroup. The only building characteristic that was significantly associated with an increase in *Legionella* concentration from either Pre-flush and Flush samples was temperature at the tap, and this was primarily driven by samples from City B and City D (Appendix A). Interestingly, City A was the only city to have an association between increasing building age and an increasing concentration of *Legionella* in both the Pre-flush and Flush samples (Figure 5). City A had the greatest mean building age compared to the other cities sampled in this study (Table 3). Other studies have shown an association with system/building age and risk of colonization [30]. For City A, no renovation was recorded, but for City C and D several buildings were renovated recently, and this might contribute to the weak or inverse association between building age and concentration (CFU/L) for those cities (Figure 5). Multiple physical, chemical, and biological factors are known to influence the growth of *Legionella* in hot water systems, but this study was limited to assessing only readily accessible building characteristics.

### 4.7. Limitations

Although the study was limited by the small number of installations investigated and the fact that strain typing was only conducted on a portion of isolates, the results showed clear differences in colonization rates, levels, and types of LP between the cities. Because temporal samples were not collected, we cannot assess whether the results presented are representative of the typical *Legionella* concentrations present in the assessed plumbing systems or temporal variations that might occur. The apartment buildings sampled in this study were chosen to reflect a range of sizes and geographic spread across each city’s potable water system; however, it is likely that they do not appropriately reflect the full range of water present in each city. Multiple buildings were sampled from each city to aid in our understanding of city-wide trends. While we could not test the DHW for water quality markers, drinking water in Denmark is known for its high levels of dissolved calcium and magnesium (i.e., hardness) [54]. Generally, cities A, B, and C have medium to hard water (12 ≤ 18° dH) while city D has soft to medium-hard water (8 ≤ 12° dH) [55]. This might be a factor worthy of investigation in future studies.

It is not known if the increase in the number of cases reported in the last seven years can be explained by an increase in colonization rates and levels of LP in DHW systems. One practical measure reducing the incidence of LD in Denmark would be to raise the temperatures for the hot water systems to at least 53–55 °C, measured at the most remote tap, but this might conflict with energy-saving measures and may increase limescale precipitation. More than half of the DHW systems sampled in this study had temperatures below the recommended minimum temperature to control and prevent growth of *Legionella* and other bacteria (50 °C). The temperature was measured after one minute of flush, and may not in all cases represent the maximum temperature that could be obtained after prolonged flushing, but a temperature of at least 50 °C should be reached within one minute according to recommendations [29,56,57]. In the Danish Code of Practice for domestic water supply systems, guidelines are given for pipe length and diameter that should ensure hot water (≥50 °C) much faster than after a one minute flush [58].

As culture is a rather insensitive method, less than half of PCR positive cases are culture-confirmed, and matching with environmental isolates is difficult (e.g., by direct SBT on clinical samples, not applied in this study) or impossible. Despite Denmark´s high rate of culture-confirmed cases (around 40%) compared to other countries, the lack of culture-positive cases, especially in the low incidence cities in this study, hampered the total overview of sero/subgroups and STs causing LP infections.

## 5. Conclusions

This study revealed high colonization rates of *Legionella*, and a large variety of SG and ST detected within DHW systems. No single factor explored in this study could cohesively explain the differences in incidence rate between these four Danish cities. Rather, a combination of factors was hypothesized to influence the incidence rate of LD in each city, including sequence type and serogroup distribution, colonization rate, concentration of *Legionella* in Pre-flush and Flush samples, and potentially building characteristics such as water temperature measured at the point of use. The large difference in notification rates between high- and low-incidence cities was not directly correlated with the general colonization rate and concentrations of *Legionella* of the DHW systems. This is exemplified by Cities A and C, which shared similar levels of *Legionella* in the systems but had a fivefold difference in the reported incidence.

More than 80% of LD cases in Denmark are diagnosed by PCR [3,20], likely increasing the relative prevalence of non-SG1 cases, which makes comparison with other countries challenging as many countries rely on *L. pneumophila* urinary antigen tests that mainly detect SG1 and may underestimate the real disease burden of *Legionella* [59,60,61]. A combination of high DHW colonization rates, high test intensity (on average, 300 to 400 persons per 100,000 are investigated each year) and use of PCR as the main diagnostic method for LD might explain the high notification rate of LD in Denmark [3].

The results are important for potential prevention methods centered on potential water supply sources, rather than cooling towers or other common outbreak settings. This study indicates a sporadic presence of LP SG1 MAb 3/1-P in DHW systems in Denmark, as it was not detected in any of the samples investigated, despite previous detection during case follow-up sampling from household DHW systems of six of the ten MAb 3/1-P ST1/ST42 cases in City A (Table 2). City B also experienced a high proportion of SG1 MAb 3/1-P cases (*n* = 13) (Table 2), but seven of the eight ST9 cases occurred over a short period of time and the type was not detected in any of the DHW systems investigated, indicating another niche infection route for this ST. A sporadic presence in DHW systems of *L. pneumophila* SG1 MAb 3/1-P ST1/ST42 (as documented in City A during previous case source investigations) and a more widespread occurrence of SG1 MAb 3/1-N ST1 (as in City B), together with high colonization rates and concentrations of *L. pneumophila* non-SG1, could contribute to regional differences in LD incidence in Denmark.

## Figures and Tables

**Figure 1 ijerph-19-02530-f001:**
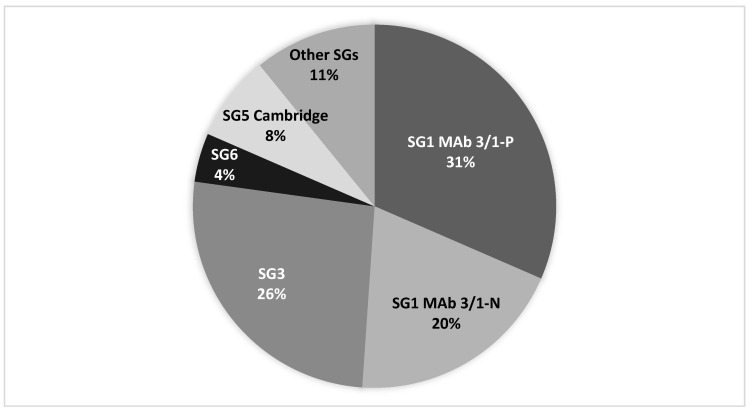
*L. pneumophila* sero/subgroup distribution for clinical isolates from 92 LD cases from the four cities (see Table 2).

**Figure 2 ijerph-19-02530-f002:**
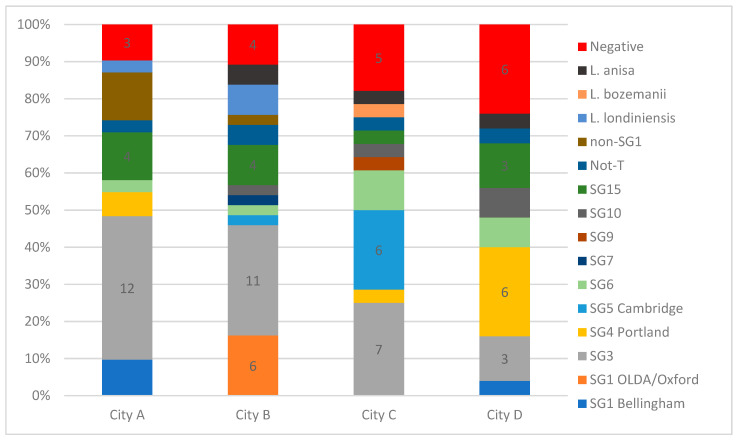
The relative proportions of *Legionella* species and *L. pneumophila* sero/subgroups detected in the systems of the four cities investigated. The number of systems with no *Legionella* detected by culture are shown in red. The number of systems with the most prevalent sero/subgroups are shown with numbers. We did not obtain non-SG1 serogroup results for five systems colonized by non-SG1; the five systems are indicated as non-SG1. In some cases, more than one (up to three) species and sero/subgroups were identified in each system; thus, the total number of species and sero/subgroup results is greater than the sum of culture-positive systems in each city.

**Figure 3 ijerph-19-02530-f003:**
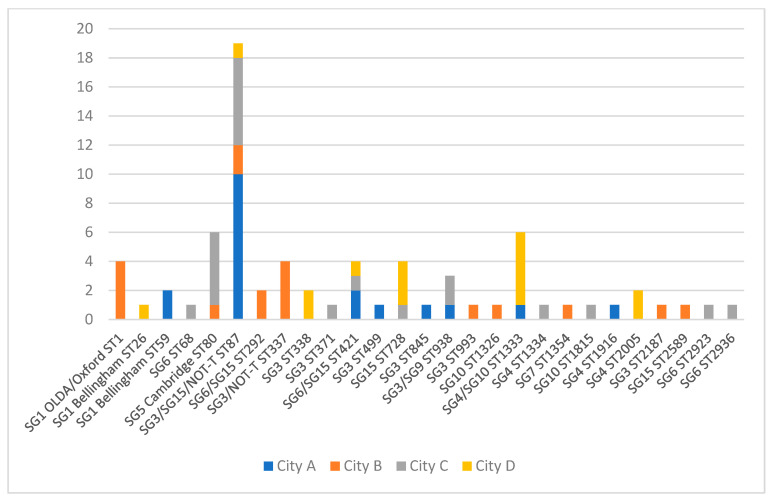
Sequence type distribution of *L. pneumophila* isolates from DHW of four cities, Denmark. The serogroups associated with each ST are indicated. NOT-T (not typeable) are isolates not reacting with monoclonal antibodies to serogroup 1 to 16.

**Figure 4 ijerph-19-02530-f004:**
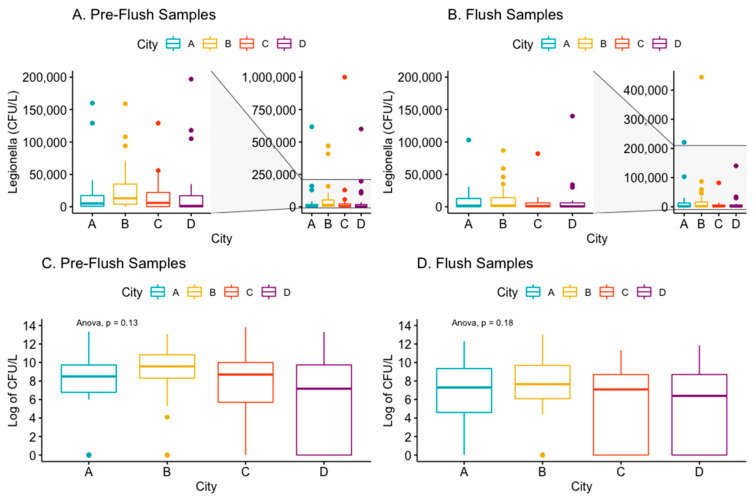
Box and whisker plots of the concentration (CFU/L) of *Legionella* (or log of *Legionella* concentration) by town in the Pre-Flush (panels **A**,**C**) and Flush (panels **B**,**D**) samples.

**Figure 5 ijerph-19-02530-f005:**
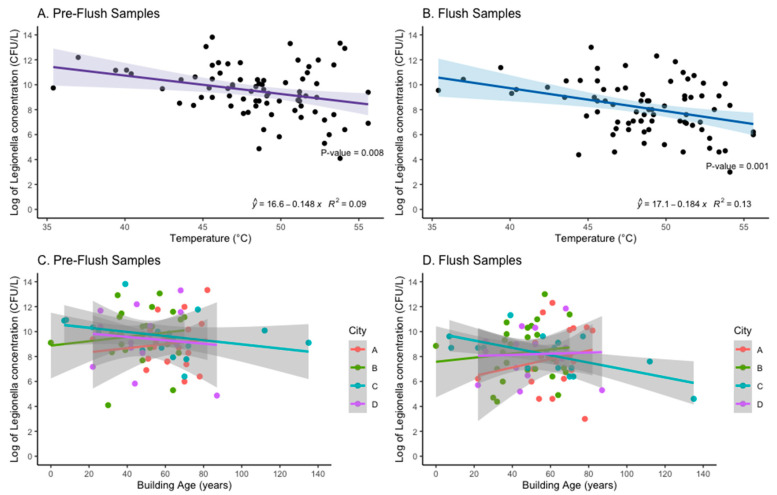
The association between log of *Legionella* concentration (CFU/L) and Temperature (°C) (Panel **A**,**B**) and Building Age (years) (Panels **C**,**D**) with line of the best fit (linear regression) and R-squared shown for Panel A and B.

**Table 1 ijerph-19-02530-t001:** Characteristics of registered (and culture-confirmed) Legionnaires´ disease cases from four cities, Denmark, 2010–2020.

City	Total No. of Cases (Culture-Confirmed)	No. of Healthcare Associated Cases (Culture-Confirmed)	Population Size (2017) × 1000	Years of Data Included	Incidence Rate * (Culture-Confirmed)	% Male (Culture-Confirmed)	Median Age Years (Culture-Confirmed)
A	46	6	64	7	10.3	46%	64
	(23)	(2)			(5.1)	(48%)	(61)
B	100	12	143	7	10.0	56%	74
	(52)	(5)			(5.2)	(56%)	(74)
C	25	1	116	11	2.0	60%	66
	(11)	(1)			(0.9)	(60%)	(65)
D	20	1	80	11	2.3	60%	73
	(6)	(1)			(0.7)	(50%)	(75)

* Incidence rates (cases per 100,000 inhabitants per year).

**Table 2 ijerph-19-02530-t002:** Typing results for clinical isolates from 92 LD cases from the four cities.

Serogroup	Subgroup	MAb 3/1	ST	City A	City B	City C	City D	Total
1	Philadelphia	POS	1	6	1	2		9
1	OLDA/Oxford	NEG	1	1	13 *			14
1	OLDA/Oxford	NEG	154		1			1
1	OLDA/Oxford	NEG	1071		1			1
1	Knoxville	POS	9		8			8
1	Knoxville	POS	1256		2			2
1	Benidorm	POS	42	4				4
1	Benidorm	POS	1806				1	1
1	Bellingham	NEG	59	1 *				1
1	Bellingham	NEG	334	1				1
1	All./France	POS	62	1	1			2
1	All./France	POS	82			2		2
1	All./France	POS	109		1			1
3		-	87	5 *	2 *	3 *		10
3		-	93	2		1	2	5
3		-	337		2 *			2
3		-	338				1 *	1
3		-	371			1		1
3		-	845		1			1
3		-	996			1		1
3		-	1609				1	1
3		-	2207	1				1
3		-	2937	1				1
4	Portland	-	1323		1			1
4	Portland	-	1535		1			1
5	Cambridge	-	80		6 *	1 *		7
6		-	728		2			2
6		-	1609		2			2
10		-	1323		1			1
10		-	1745		1			1
13		-	337		1			1
16		-	1333				1 *	1
NOT-T		-	87		1 *			1
NOT-T		-	1326		1 *			1
NOT-T		-	2227		1			1
NOT-T		-	UNK		1			1
Total				23	52	11	6	92

* ST found in water samples from each City investigated in this study; NOT-T: isolate does not belong to *L. pneumophila* serogroup 1–16.

**Table 3 ijerph-19-02530-t003:** Characteristics of *Legionella* (CFU/L) among Pre-flush and Flush samples from four cities, Denmark, 2020. Only systems with growth of *L. pneumophila* are included as positive.

	A	B	C	D	ANOVA ^†^
No. of Systems investigated (No. of water samples)	24 (48)	30 (60)	25 (50)	25 (50)	-
Proportion of systems positive for *L. pneumophila*	(21/24)	(25/30)	(18/25)	(17/25)	0.20 **
	87.5%	83.3%	72%	68%	
Pre-flush Median (range) CFU/L	5000	14,500	6000	1300	-
	(0 *–617,000)	(0 *–470,000)	(0 *–1,000,000)	(0 *–600,000)	
Flush Median (range) CFU/L	1550	2150	1200	650	-
	(0 *–221,000)	(0 *–444,000)	(0 *–82,000)	(0 *–140,000)	
Pre-flush Mean (SD) CFU/L	44,604	55,565	56,144	45,591	0.71
	(128,333)	(111,243)	(198,643)	(124,939)	
Flush Mean (SD) CFU/L	19,118	26,669	6860	9885	0.91
	(48,365)	(81,420)	(16,423)	(28,509)	
Hot Water Tank Present (%)	0%	90%	12%	12%	<0.001
Building Size Mean (SD)	49.1 (41.9)	51.9 (38.5)	41.2 (35.8)	33.2 (22.9)	0.23
Building Age (years)	57.2 (18.4)	46.2 (17.5)	51.6 (32.1)	48.7 (22.1)	0.36
Temperature Mean (SD)	50.80 (2.75)	48.2 (5.25)	48.1 (4.21)	48.5 (3.49)	0.07

* 0 = <100 CFU/L; ^†^ ANOVA calculated using log (*Legionella* concentration); ** Chi-square test.

**Table 4 ijerph-19-02530-t004:** Concentration of *Legionella* (CFU/L) as a proportion of Pre-flush and Flush samples within the indicated ranges. Only systems with growth of *L. pneumophila* are included as positive.

City	Pre-Flush * 0–100 CFU/L No. (Row %)	Flush * 0–100 CFU/L No. (Row %)	Pre-Flush ≥ 100 to ≤1000 CFU/L No. (Row %)	Flush ≥ 100 to ≤1000 CFU/L No. (Row %)	Pre-Flush > 1000 to ≤10,000 CFU/L No. (Row %)	Flush > 1000 to ≤10,000 CFU/L No. (Row %)	Pre-Flush > 10,000 CFU/L No. (Row %)	Flush > 10,000 CFU/L No. (Row %)
A	4 (17)	4 (17)	3 (13)	7 (29)	8 (33)	7 (29)	9 (38)	6 (25)
B	6 (20)	5 (17)	1 (3)	6 (20)	7 (23)	9 (30)	16 (53)	10 (33)
C	7 (28)	7 (28)	1 (4)	4 (16)	7 (28)	10 (40)	10 (40)	4 (16)
D	10 (40)	8 (32)	2 (8)	6 (24)	6 (24)	8 (32)	7 (28)	3 (12)

* 0 = <100 CFU/L.

## Data Availability

Data are available on request.

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
