# Peer review of "A Tale of Four Danish Cities: Legionella pneumophila Diversity in Domestic Hot Water and Spatial Variations in Disease Incidence"

_ijerph, 2022, doi:10.3390/ijerph19052530_

Round 1
Reviewer 1 Report
The manuscript “A tale of four Danish cities: Legionella pneumophila diversity in domestic hot water and spatial variations in disease incidence” by Uldum et al. investigated the colonization rates/levels, and sero/subgroup-ST of Legionella pneumophila in four Danish cities with different LD incidence. Variable as building age and size, temperature and presence of hot water tanks were considered.
I have found the study very interesting and here I write down some points:
- The high Lp colonization rates of DHW in all the investigated cities with a percentage of systems at high levels (>10,000 CFU/L).
- Low proportion of Pontiac strains isolated from the DHW, in spite of a discrete number of Pontiac culture-confirmed cases detected.
- High proportion of non-Pontiac strains among cases but low proportion of these strains in the investigated systems.
- The evidence that one (ST87 linked to sg3) of the two main STs found in Denmark (ST1 and ST87) were isolated in all the four cities, although with different prevalence.
I want to suggest more than minor revision any little remark.
An important consideration is that one half of cases are Lp1 and one half Lp non-sg1, these last probably detected by PCR, and the DHW are prevalently contaminated by Lp-nonsg1. I have found interesting the widespread Legionella colonization of domestic water system, scarcely investigated in other countries. This could be emphasized.
Most of the cases diagnosed by PCR remain without a source of infection because of the lack of a matching with environmental strains. This also could be remark.
The sg3, one of the most abundant sg in Denmark, was found associated to ST87 in all the four cities. Are there sg3 associated to other STs? Were there cases due to sg3 ST87 for which an environmental correlation was determined, and specifically with strains isolated in the patient’s home?
Only if available, a brief description of the characteristics of susceptibility of the LD cases could be helpful to understand the high incidence of “domestic” cases, where non-Pontiac strains were isolated.
Author Response
ijerph-1588731
“A tale of four Danish cities: Legionella pneumophila diversity in domestic hot water and spatial variations in disease incidence”
Responses to reviewer 1 comments
Reviewer 1
1) An important consideration is that one half of cases are Lp1 and one half Lp non-sg1, these last probably detected by PCR, and the DHW are prevalently contaminated by Lp-nonsg1. I have found interesting the widespread Legionella colonization of domestic water system, scarcely investigated in other countries. This could be emphasized.
Response 1: Thank you for the comment and suggestion
Comment to “…one half of cases are Lp1 and one half Lp non-sg1, these last probably detected by PCR”:
The majority of all cases in Denmark are diagnosed by PCR irrespective of the serogroup. At SSI we receive most of the samples found PCR positive at local laboratories in Denmark, at SSI we investigate the samples both for Legionella spp. (16S), L. pneumophila (mip) and serogroup 1 (wzm). The proportion between SG1 and non-SG1 varies a little from year to year, the last two years almost 59 % were SG1 by PCR, by culture however, the SG1 strains only accounted for 48% of isolates (all PCR positive samples are cultured – less than half will grow Legionella). This correlates with a generally higher DNA load in samples from non-SG1 cases and a higher cultivability than SG1. In the present study, travel-associated cases were exclude; ~90% of those cases (normally 20-25% of all cases) are infected by SG1 Mab 3/1 positive strains. In addition, few local laboratories do also culture of all PCR positive samples.
Comment to “..scarcely investigated in other countries. This could be emphasized.”
In the Discussion section, 5.3. Discussion of SG MAb 3/1 and non-SG1, we have added a discussion on this, primarily addressing SG3.
2) Most of the cases diagnosed by PCR remain without a source of infection because of the lack of a matching with environmental strains. This also could be remark.
Response 2: Thank you for this comment as it is important to address. In Denmark, we strive to culture most cases, however more than half are either only PCR or/and urinary antigen positive, in these cases we cannot directly do matching with environmental strains. Although we have a high positive culture rate of around 40%, an epidemiological link is only found for half of the investigated cases. In most cases the matching isolate is detected in the residential water system. For PCR positive patients there are however the opportunity (in contrast to urinary antigen positive patients) to do direct SBT (Sequence Typing) on the DNA extracted from the clinical sample and a ST matching is possible. This is however, only performed in few cases e.g. if nosocomial infection is suspected. The method is only possible if the DNA load is rather high, in some cases only a partial SBT profile is obtained – that can however also be helpful. As we have many cases of ST1 and ST87 which is widespread, in some areas at least, a matching only on ST can be doubtful, and further SNP analysis (as we have the whole genome sequenced for all clinical isolates) are needed and used with some success – this is mostly applied in nosocomial, institutional-, hotel-associated and cluster cases. The workload and costs for single community acquired case is too high.
In the Discussion section, we have added the following in
5.7. Limitations
As culture is a rather insensitive method, less than half of PCR positive cases are culture-confirmed and matching with environmental isolates is difficult (e.g. by direct SBT on clinical samples, not applied in this study) or impossible. Despite Denmark´s high rate of culture-confirmed cases (around 40%) compared to other countries, the lack of culture positive-cases, especially in the low incidence cities in this study, hampered the total overview of sero-/subgroups and STs causing LP infections.
3) The sg3, one of the most abundant sg in Denmark, was found associated to ST87 in all the four cities. Are there sg3 associated to other STs? Were there cases due to sg3 ST87 for which an environmental correlation was determined, and specifically with strains isolated in the patient’s home?
Response 3: Thank you for this question. SG3 is associated with several STs; as seen in Table 2 SG3 has been associated with ten different STs in the four cities. ST93 is next to ST87 the most prevalent SG3 ST detected in clinical samples, and ST93 is the third most prevalent ST after ST1 and ST87 – accounting for around 5% of cases. Altogether we have identified more than 22 different STs among clinical and environmental SG3 isolates in Denmark during the last five years. ST87, ST93, ST337, ST338, ST845, ST938, ST993 are the most common both in clinical and environmental samples (primarily DHW systems).
For clinical ST87 cases in this study five cases were investigated and the matching ST was found in patients DHW system. For SG3 ST93 a matching isolate was found in two cases, the total number of investigations is unknown for us.
We have added the following in the Discussion section 5.3. Discussion of SG1 MAb 3/1-N and non-SG1
Of the 24 SG3 cases from the four cities (Table 2) it is known that the corresponding serogroup and ST were detected in DHW samples for seven of the cases (ST87 in five and ST93 in two cases). SG3 ST93 is the third most common ST (after ST1 and ST87) detected in clinical samples in Denmark and accounts for approximately 5% of cases (the same proportion as found in the four cities (Table 2)). This ST was not detected in water samples from this survey but the ST (SBT profile, 3,10,1,28,14,9,13) is relatively closely related to ST87, sharing four common ST loci.
The relatedness of STs found in the four cities is further discussed with results from SNP analysis – see below response to reviewer 2.
4) Only if available, a brief description of the characteristics of susceptibility of the LD cases could be helpful to understand the high incidence of “domestic” cases, where non-Pontiac strains were isolated.
Response 4: Thank you for the suggestion. The only known susceptibility marker known for patients in this study is age. We have calculated the mean age for SG1 MAb 3/1-P and compared to SG1 MAb 3/1-N and non-SG1 cases and found a significant higher age for the last group, that especially is supposed to have acquired the infection at home. We have however not discussed it further in this paper, as we do not know the general health conditions or other pre-disposing factors for the patients.
We did additions in following sections – please see revised manuscript
Material and Methods
3.8. Assessing the association between age of culture positive cases and infection with MAb 3/1 positive strains vs. MAb 3/1 negative and non-SG1 strains
Results
4.1. Incidences of LD cases and culture-confirmed LD cases in the four cities and types detected in clinical samples

Reviewer 2 Report
In their manuscript Uldum et al compare L. pneumophila population (both clinical and environmental) in four Danish cities, two (A and B) with the high incidence rate of Legionnaires’ disease (LD) cases and two (C and D) with the low incidence rate of LD cases. It is an informative paper that reveals rather unusual L. pneumophila population in Denmark with Lp sg1 Mab3/1-positive strains representing only 31% of the population of clinical Lp isolates, whereas Lp sg3 causes 26% of the LD cases. Among the environmental isolates, the Lp sg 3 is found in all 4 cities, while the Lp sg1 Mab3/1-positive strains were not identified in any of the cities’ Domestic Hot Water systems.
Specific comments:
- I recommend not to use the term “Pontiac strains” to describe Mab3/1 – positive Lp1 strains. I do not think that this is a commonly used term. The paper [9] (Thürmer et al, PCR-Based “serotyping” of Legionella Pneumophila. J Med Microbiol 2009, 58, 588-595), cited by the authors when they introduce the term “Pontiac strains” (page 2, first paragraph), does not appear to use the “Pontiac strains” term either. The authors do not provide any explanation why this specific term was selected. It could be confused with the historical US L. pneumophila sg1 Pontiac strain, with the Pontiac Fever disease, or with Lp1 Mab subgroups, many of which are named after geographical locations (e.g. Knoxville, Philadelphia or Oxford). I would recommend do use either “lag1-positive” or “Mab3/1-positive” term instead.
- Materials and Methods: for the 3.1 “Culture -confirmed LD case assessment in four Danish cities”, it would be appropriate to reference Table 1. If the authors do not wish to reference Table 1 here, then they should change the numbering order for the tables, since the first mentioning of any tables appears to be of Table 2 in Methods 3.5 “Whole-genome sequencing and sequence type determination”. The table that is referenced first should be numbered “Table 1.”
- Results: 4.1 “Incidences of LD cases and culture-confirmed LD cases in the four cities and types detected in clinical samples”; page 4, first line: there should be 191 LD cases, not 190. Based on the data in Table 1, the sum of cases in the second column is 46 + 100 + 25 + 20 = 191.
- Table 1, third column: I assume that the second number for each row represents the number of the culture confirmed cases. If this is correct, then the “(culture confirmed)” should be added to the header of the third column and the second number in each row should be in parenthesis.
- Do any of the sequence types identified for the clinical and environmental isolates closely relate to each other or form clonal complexes? I am especially interested in relationship between STs of non-sg1 L. pneumophila that belong to the same serogroup, for example sg3.
- It could be useful to show comparison of sequence types and serogroups of clinical vs environmental isolates from the same city in the same figure or table, so one can easily see whether the city-specific clinical strains were also identified in the city’s DHW. It is challenging to go back and forth between Table 2 and Figure 3 to make this comparison.
- Discussion: I wonder if the authors could discuss the possible reasons for the unusually high proportion of Lp sg3 among both environmental and clinical isolates. I do not believe it is typical for other European countries, but there was at least one publication about the prevalence of Lp sg3 in Israel (Yarom et al, Legionella pneumophila serogroup 3 prevalence in drinking water survey in Israel (2003 – 2007), WSTWS 2010.). Are there any factors shared by both Denmark and Israel that cause the predominance the Lp3 strains?
Author Response
ijerph-1588731
“A tale of four Danish cities: Legionella pneumophila diversity in domestic hot water and spatial variations in disease incidence”
Responses to reviewer 2 comments
Reviewer 2
- I recommend not to use the term “Pontiac strains” to describe Mab3/1 – positive Lp1 strains. I do not think that this is a commonly used term. The paper [9] (Thürmer et al, PCR-Based “serotyping” of Legionella Pneumophila. J Med Microbiol 2009, 58, 588-595), cited by the authors when they introduce the term “Pontiac strains” (page 2, first paragraph), does not appear to use the “Pontiac strains” term either. The authors do not provide any explanation why this specific term was selected. It could be confused with the historical US L. pneumophila sg1 Pontiac strain, with the Pontiac Fever disease, or with Lp1 Mab subgroups, many of which are named after geographical locations (e.g. Knoxville, Philadelphia or Oxford). I would recommend do use either “lag1-positive” or “Mab3/1-positive” term instead.
Response 1. Thank you for the comment and suggestions regarding the use of Pontiac, non-Pontiac. We agree that the name is outdated and the name Pontiac can be confused with Pontiac fever. The name Pontiac refers to the Pontiac strain (ST62) (that caused the Pontiac outbreak) which is MAb 3/1 (Dresden panel) or Mab 2 (International Panel) positive and has named the whole group (see below). Some use the term in Europe – but not widely used, the name Pontiac has however the advantage of being a generic name not dependent on the method how this functionality is detected, it could either be by MAb 3/1 or MAb 2 (or other Mab´s) or by specific PCR primers, or by DNA sequencing as WGS. In this study, however, all strains were identified by serology with the MAb 3/1. Strains can be lag-1 “positive” but harboring a non-functional gene due to either truncation, insertions or mutations, then we do not prefer to characterize the group as lag-1 positive. The use of the name Pontiac can be traced back to the study by Watkins et al. Legionella pneumophila serogroup 1 subgrouping by monoclonal antibodies – an epidemiological tool. J Hyg Camb 1985, 95, 211-216, the name has stuck to the classification by the Dresden Panel of monoclonal antibodies for SG1 subtyping.
The reference [9] refers to the association between lag-1 gene functionality and MAb 3/1 reactivity, and we agree it does not refer to the use of the term Pontiac, and the sentence has been changed to “The MAb 3/1 recognize an LPS epitope encoded by a functional lag-1 gene (an O-acetyltransferase) [9].”
The term Pontiac and non-Pontic has been deleted throughout the paper and replaced by MAb 3/1 positive or MAb 3/1 negative – abbreviated to MAb 3/1-P and MAb 3/1-N. We also changed the abbreviation of monoclonal antibodies to MAb instead of Mab.
- Materials and Methods: for the 3.1 “Culture -confirmed LD case assessment in four Danish cities”, it would be appropriate to reference Table 1. If the authors do not wish to reference Table 1 here, then they should change the numbering order for the tables, since the first mentioning of any tables appears to be of Table 2 in Methods 3.5 “Whole-genome sequencing and sequence type determination”. The table that is referenced first should be numbered “Table 1.”
Response 2: Thank you for this suggestion. We have included a reference to Table 1 in Materials and Methods 3.1 Culture-confirmed LD case assessment in four Danish cities.
- Results: 4.1 “Incidences of LD cases and culture-confirmed LD cases in the four cities and types detected in clinical samples”; page 4, first line: there should be 191 LD cases, not 190. Based on the data in Table 1, the sum of cases in the second column is 46 + 100 + 25 + 20 = 191.
Response 3: Thank you for noticing this, 190 is corrected to 191 in Results 4.1 first line.
- Table 1, third column: I assume that the second number for each row represents the number of the culture confirmed cases. If this is correct, then the “(culture confirmed)” should be added to the header of the third column and the second number in each row should be in parenthesis.
Response 4: Thank you for this suggestion. It is correct that the second line in the third column in Table 1 is culture-confirmed cases; (culture confirmed) has been added to the header.
- Do any of the sequence types identified for the clinical and environmental isolates closely relate to each other or form clonal complexes? I am especially interested in relationship between STs of non-sg1 L. pneumophila that belong to the same serogroup, for example sg3.
Responses 5: We are happy to address this question. SNP analysis was something we considered to include in the paper, however decided not to do as it could be considered as a little out of scope, and a little complicated. We have now constructed phylogenetic trees based on SNP analysis for isolates from the four cities, both for unpurged (recombination not removed) and purged (recombination removed) results. In L. pneumophila horizontal recombination is a major driver for evolution and divergence, whereas often only few SNP differences are found when recombination is removed, it is why we have included both trees. The trees are in the Appendix Figure C and D. The field is complex and our presentation is not exhaustive. We hope that we have addressed your questions by adding this.
Method has been added:
3.5. Whole-genome sequencing, sequence type determination, and SNP analysis
Results has been added:
4.6. Single nucleotide polymorphism (SNP) analysis
Discussion has been added:
5.5. Discussion of SNP analysis
Two Figures is added to the supplementary material (Appendix, Figure C and D) with unpurged and purged SNP phylogenetic trees.
- It could be useful to show comparison of sequence types and serogroups of clinical vs environmental isolates from the same city in the same figure or table, so one can easily see whether the city-specific clinical strains were also identified in the city’s DHW. It is challenging to go back and forth between Table 2 and Figure 3 to make this comparison.
Responses 6: Thank you for this suggestion. We did however in Table 2 mark* all STs that both were identified in clinical isolates and in the DHW system in the corresponding city. It is not in all cases there is correspondence between the serogroup detected in clinical isolate and in environmental isolate, e.g. for ST1333 detected in City C where the clinical isolate is typed as SG16, whereas the environmental isolates were typed as SG4 and 10. The serogroup for this ST is variable – and it is difficult to distinguish SG4 and 16 for some isolates as there are some cross-reactions. We consider a ST result as more robust and objective than serotyping result. We hope that this marking in Table 2 is sufficient to identify correspondence between clinical and environmental isolates.
- Discussion: I wonder if the authors could discuss the possible reasons for the unusually high proportion of Lp sg3 among both environmental and clinical isolates. I do not believe it is typical for other European countries, but there was at least one publication about the prevalence of Lp sg3 in Israel (Yarom et al, Legionella pneumophila serogroup 3 prevalence in drinking water survey in Israel (2003 – 2007), WSTWS 2010.). Are there any factors shared by both Denmark and Israel that cause the predominance the Lp3 strains?
Responses 7: We agree that this should be discussed more thoroughly. Unfortunately, it is common for similar studies to group SG analysis into SG1 and SG2-14 (or SG2-16, or non-SG1) making direct comparisons of our SG3 results challenging.
According to Data from ECDC, Legionnaires´ Disease, Annual Epidemiological Report for 2019, AER Legionnaires 2019 (europa.eu) altogether 35 SG3 cases were culture confirmed in EU/EEA 2019 making SG3 the second most prevalent serogroup (3%), however, 25 of the cases were Danish. Only relatively few cases were culture confirmed in Europe (10%) whereas the culture positive rate was 40% in Denmark in 2019. Nevertheless, SG3 seem to be a special problem in Denmark although there probably are some underdiagnosing of SG3 and other non-SG1 cases in Europe. The main method for diagnosing LD in Europe was/is urinary antigen test (90% of cases) detecting only SG1. It is the opposite in Denmark where 90% are diagnosed by PCR that can detect all serogroups and the sample can be cultured. Therefore, it is hard to compare the Danish prevalence of SG3 in clinical isolates with other countries. An older Pan-European study by Helbig et al. 2002 (at a time where the use of urinary antigen test not was as prevalent as it is today and the culture rate was higher (17%)), showed that 21% of cases were non-SG1 with SG3 and SG6 dominating. It was also shown that the proportion of MAb 3/1 negative strains was higher in the Scandinavian countries (48.7%, community acquired) than in the Mediterranean and UK (18.6 and 12% respectively).
We have added a discussion about SG3 in the following section:
5.3. Discussion of SG1 MAb 3/1 negative and non-SG1
